# Neural correlates of episodic memory modulated by temporally delayed rewards

**Jungsun Yoo[1¤], Seokyoung Min[1], Seung-Koo Lee[2,3], Sanghoon Han[1]***

**1** Department of Psychology, Yonsei University, Seoul, Republic of Korea, **2** Department of Radiology, Yonsei University College of Medicine, Seoul, Republic of Korea, **3** Integrated Neurocognitive Functional Imaging Center, Yonsei University, Seoul, Republic of Korea

¤ Current address: Department of Cognitive Sciences, University of California, Irvine, California, United States of America
* sanghoon.han@yonsei.ac.kr

**Data Availability Statement:** The data used in this study are available on Figshare (DOI: 10.6084/m9. figshare.13285928.v1) and also included as a Supporting Information file.

## Abstract

When a stimulus is associated with an external reward, its chance of being consolidated into long-term memory is boosted via dopaminergic facilitation of long-term potentiation in the hippocampus. Given that higher temporal distance (TD) has been found to discount the subjective value of a reward, we hypothesized that memory performance associated with a more immediate reward will result in better memory performance. We tested this hypothesis by measuring both behavioral memory performance and brain activation using functional magnetic resonance imaging (fMRI) during memory encoding and retrieval tasks. Contrary to our hypothesis, both behavioral and fMRI results suggest that the TD of rewards might enhance the chance of the associated stimulus being remembered. The fMRI data demonstrate that the lateral prefrontal cortex, which shows encoding-related activation proportional to the TD, is reactivated when searching for regions that show activation proportional to the TD during retrieval. This is not surprising given that this region is not only activated to discriminate between future vs. immediate rewards, it is also a part of the retrieval-success network. These results provide support for the conclusion that the encoding-retrieval overlap provoked as the rewards are more delayed may lead to better memory performance of the items associated with the rewards.

## Introduction

Episodic long-term memory is crucial to our daily lives given that our behavior and decisions are based on past experience. The core of episodic memory formation in the brain is thought to be the medial temporal lobe (MTL), which comprises the hippocampus (HC) and the surrounding gyri [1]. In addition, various studies suggest that successful memory formation does not solely depend on MTL functions but also relies on other cortical regions such as the prefrontal cortex (PFC), the premotor cortex, or the posterior parietal cortex (PPC) [2].

However, not all events benefit from these regions to be transformed into long-term memory. To enhance efficiency in memory storage, salient stimuli–either rewarding or aversive–are better remembered than other neutral items [3]. Upon encountering rewarding stimuli, dopamine neurons in the substantia nigra and the ventral tegmental area (VTA), two regions

**Funding:** This research was supported by the Brain Research Program through the National Research Foundation of Korea(NRF) funded by the Ministry of Science and ICT (2017M3C7A1029485) and partially supported by the National Research Foundation of Korea(NRF) grant funded by the Korea government (NRF-2019R1A2C1007399). The funders had no role in study design, data collection and analysis, decision to publish, or preparation of the manuscript.

**Competing interests:** The authors have declared that no competing interests exist.

that produce most of the dopamine in the brain, are activated [4]. Dopamine is known to facilitate consolidation of long-term memory via direct projections of the VTA to the HC [5]. This reward-facilitated memory consolidation is observed 24 hours after the initial learning [6] through late long-term potentiation in the HC [7,8]. Various studies have confirmed the effect of reward-facilitated episodic-memory encoding in the human brain. One study revealed that stimuli that elicit greater activation in the dopaminergic midbrain areas are more likely to be recollected three weeks later [9], while another study reported that a high-value reward preceding incidental encoding facilitates memory formation [10]. In summary, a high-value stimulus has a higher chance of being consolidated into the long-term memory (for a review, see [11]).

The subjective value of a reward has been found to vary according to various contexts such as the amount of effort needed for obtaining the reward, probability associated with the reward, or delay before the reward is delivered. Effort discounting refers to the tendency of choosing a reward option associated with less over more required effort. Activation of the nucleus accumbens was negatively correlated with effort demand [12]. Likewise, with a fixed objective value of a reward, its subjective value decreases if the probability of obtaining it decreases [13]. Temporal discounting of reward refers to the tendency of individuals to devalue the reward if its delivery is delayed [14], often resulting in choosing smaller sooner over larger later rewards. There have been various attempts to capture this phenomenon into mathematical equations, and a model using hyperbolic function has provided the most parsimonious account so far [15,16]. A neuroimaging study has corroborated this by finding that the subjective value of delayed rewards derived from the hyperbolic model is represented in the ventral striatum, the mPFC, and the posterior cingulate cortex [17].

Given that a TD of a reward reduces the activation of dopamine neurons as if a reward of a smaller magnitude has been expected [18–20], one could link this with reward-motivated memory-encoding by questioning whether devalued rewards due to a higher TD act like a small-magnitude reward in memory encoding. With regard to reward contexts other than TD, a study examined effects of reward uncertainty and magnitude on episodic memory and found that only reward outcome, but not reward uncertainty, affects episodic memory [21]. However, to our knowledge, no study attempted to test whether temporally discounted rewards differentially affect episodic memory formation. Therefore, in this study, we aim to investigate whether temporal discounting of reward is reflected in reward-motivated episodic memory encoding, and if so, whether it involves differential HC-VTA activity.

In this study, we hypothesize that the more temporally distant a reward is, the less likely the stimulus associated with it will be remembered. To test this, we devised an incidental memory-encoding task in which participants encode scene images preceded by a reward cue indicating when the reward will be delivered should the given trial be correct. We assume that the effect of reward modulation by TD will not be seen pre-consolidation but after consolidation, since dopaminergic facilitation of episodic memory is observed after 24 hours [6]. To validate this assumption, we divide the retrieval phase into two phases–pre-consolidation (referred to as 'recent retrieval' or 15 minutes after encoding) and post-consolidation (referred to as 'remote retrieval' or one week after encoding)–and acquire functional magnetic resonance imaging (fMRI) images from both retrieval phases to directly test whether the effect of temporally-discounted reward depends on dopamine-facilitated consolidation.

## Materials and methods

### Participants and materials

N = 26 volunteers participated in the study (12 women). Two subjects were excluded from the study due to failure to participate on the second day of the experiment. Two additional

participants were excluded since they had at least one condition without a usable regressor for General linear modeling (GLM) analyses of fMRI data. The volunteers' age ranged from 19 to 31, and their mean age was 25. Before study begin, all participants were screened if they matched the inclusion criteria of having no history of psychological or neurological disorder, not being pregnant, having no claustrophobia or tinnitus, and being right-handed. Only volunteers who met all the criteria were enrolled. All participants gave their informed written consent according to a document approved by the Institutional Review Board of Yonsei University. We acquired 22 complete fMRI datasets (244 trials of encoding, 404 trials of retrieval), and two participants' data was partially lost due to malfunction of MRI scanner (1st participant: loss of 41 trials during first retrieval phase; 2nd participant: loss of 114 trials during first retrieval phase, and this participant's data was not used for any analyses). The sample size of this study was determined by referring to the sample size of top-cited studies published during 2017–2018, which is 23–24 subjects [22]. Also, we note that our final sample size of 22 subjects falls within the range of recently published functional neuroimaging studies on episodic memory encoding and/or retrieval [23–25].

The experiment and data collection were performed via Cogent 2000 (Wellcome Trust Centre for Neuroimaging) based on MATLAB (The MathWorks; Natick, MA). For scene stimuli, 405 scene stimuli were sampled from a preexisting image database [26] or from online royalty-free images. All images were adjusted to a size of 92.1 mm x 92.1 mm and a resolution of 72 pixels. The order of visual stimuli presentation during the encoding and retrieval phase as well as the locations where the binary response options appear (left/right) were randomized. When the participants were placed inside the MRI scanner, visual stimuli were presented to them via MR-compatible goggles and their responses were recorded using an MR-compatible button-box.

## Experimental paradigm

We devised a novel experiment which crosses reward with TD during memory encoding. The experiment consisted of three main sections, which were a scene-encoding task, a surprise memory-retrieval task, and an intertemporal decision-making task. Subjects performed these tasks while they were scanned for their brain activity inside an MRI scanner. The experiment took place for two days. The first day encompassed encoding and half of the retrieval task. The second day, one week later, comprised the remaining half of the retrieval and the intertemporal choice task (Fig 1). In the encoding task, participants classified whether a given scene was indoors or outdoors and were rewarded if they made the correct classification. The novel part of this classification task is that each trial is preceded by a cue indicating when the reward will be delivered should the following classification be correct, and there were four possible TDs: 0 days (same day), 1 day, 7 days, and 28 days. The reward magnitude was uniform across all trials, resulting in 5,000 Korean Won (KRW) when summed. For retrieval, scenes presented during the encoding task intermixed with new scenes (foils) were shown to the participants who decided whether they had seen the scene in the encoding task (old/new) and how confident they were about the decision (sure/guess). Finally, to estimate participants' discount rate, the intertemporal choice task described in [17] was conducted, except that the reward was hypothetical in our study. This deviation can be ignored since no difference has been observed between real and hypothetical rewards in temporal discounting tasks [27].

## Experimental procedure

Upon arrival, participants were first asked to fill out the consent form and behavioral avoidance/inhibition scales for a measure of reward-sensitivity [28]. Then they were instructed about the scene-classification task, including the possibility of receiving an additional reward of up to 5,000

## Encoding Task

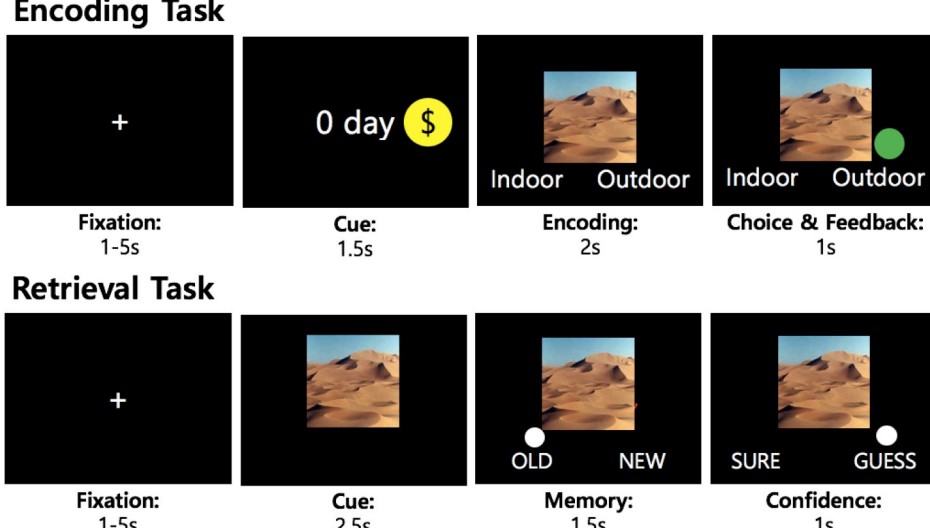

**Fig 1. Experimental paradigm.** Each trial in encoding and retrieval task was preceded with a jittered fixation. In the encoding task, correct response was followed by a green dot while incorrect response resulted in a red dot.

KRW depending on their classification accuracy and four possible time points when the reward will be delivered to them. Since the retrieval following encoding was intended to be a surprise memory test, participants were only informed that they will perform a simple cognitive task after the encoding phase. Then they were put inside the MRI scanner and a six-minute pre-encoding resting run was performed. After pre-encoding resting, participants performed six runs of encoding (scene-classification) task, with 44 items per each run, resulting in 264 scenes in total. Each condition (reward delivery after 0/1/7/28 day(s)) was comprised of 66 items. Then the post-encoding resting phase followed. After resting, participants were informed that the aforementioned 'cognitive task' is actually a recognition task and they performed the first half of the retrieval task inside the scanner, consisting of 202 items divided into five runs. After leaving the scanner, the participants were paid for the reward they earned during the encoding task.

Participants returned a week later and resumed the experiment by performing the intertemporal decision-making task three times. They also answered the Zimbardo Time Perspective Inventory (ZTPI) and an abbreviated 9-item form of Raven Standard Progressive Matrices (RSPM) [29,30] as a measure of their future orientation and intelligence, respectively. However, data collected from ZTPI and RSPM were not used for this study. Then the participants were positioned in the MRI scanner and first performed the identical intertemporal decision-making task which they performed outside the scanner. Afterward, the remaining half of the retrieval task took place, also consisting of 202 items distributed into five blocks. Finally, the structural images of participants (T1- and T2-weighted images) were acquired. The experiment terminated as the subjects were paid the rest of their participation reimbursement and provided their signature for receipt. Note that the bonus was bank-transferred according to each TD, so when returning to the experiment on the second day the participants had already received bonus for 0-days, 1-day and 7-days condition.

### Behavioral data analyses

Participants' task-specific performance during the encoding task was assessed by analyzing the response time (RT) and accuracy during scene classification. This analysis ensured that performance during the encoding task per se does not lead to differential memory performance

according to conditions in the retrieval task. For retrieval tasks, participant's memory performance and RT were assessed. First, participants' memory performance for each condition was analyzed by calculating the corrected recognition (CR), which is a model-free memory measure calculating hit minus false alarm rate ('old' responses to targets minus 'old' responses to foils). Based on the previous finding that the reward-motivated subsequent memory effect (SME) is reflected in high-confidence responses [10], we additionally calculated high-confidence hits corrected for high-confidence false alarm rate. The CRs of participants were submitted to 2x4 repeated-measures analyses of variance (ANOVA). Second, participants' RT was assessed for retrieval tasks to ensure that the results of the retrieval tasks are not explainable via RT.

Each participant's discount rate was estimated based on the results of the delay-discounting decision-making task. A logistic function was fit to choices within each delay ($D$) to determine the indifference point, which refers to the amount of money which yields 50% probability of choosing either option. The indifference point for each delay was used to calculate the discounted value (DV):

$$DV = \frac{\text{magnitude of immediate reward (\$10)}}{\text{indifference point}}$$

Then, the DVs as a function of delays were fitted by a hyperbolic function [31], where

$$DV = \frac{1}{1 + k * D}$$

The estimated free parameter $k$ was used as a measure for each individual's impulsivity, where higher $k$ leads to steeper hyperbolic curve or higher impulsivity. The purpose of deriving each participant's $k$ was to estimate the subjective value of delayed rewards in the encoding session tailored to each participant. This was done by multiplying objective numerical value for each trial (5,000 KRW/264 trials = 18.9 KRW) by the respective DV of each delay of each participant (18.9 KRW * DV). The obtained subjective value pertaining to each delay was entered as a parametric modulator for GLM of functional images acquired during retrieval runs.

## Neuroimaging data acquisition and analyses

**MRI data acquisition.** The MRI data for this study was acquired with a Phillips 3T MRI scanner using a 32-channel head coil. All functional images were T2*-weighted echo-planar images with repetition time of 0.8 s, voxel size of 2.4 x 2.4 x 2.4 mm, multiband factor of 6, and 60 slices in an ascending order, aligned to the anterior commissure–posterior commissure axis. The first ten images of each run were discarded for magnetic stabilization. A full dataset comprised of 500 volumes each for the pre- and post-encoding resting phase, 2520 (6 runs * 420 volumes per run) volumes for the encoding phase, and 3900 (2 phases * 5 runs * 390 volumes per run) volumes for the retrieval phase. For structural images, a T1-weighted MPRAGE (1 mm isotropic voxels) and a T2-weighted image were acquired. The T2 structural images were not used for any analyses in this study.

**Preprocessing.** The acquired functional images were first unwarped using the Topup toolbox of FSL software (FMRIB, Oxford, UK) to correct for possible distortions due to magnetic field inhomogeneity [32]. Then the differences in the slice acquisition time for the functional images were corrected. Afterwards, the functional images were realigned to the first image of each run to correct for movement, and a mean functional volume was created to use as a representative for coregistration of functional and structural (T1) images. The coregistered structural image was segmented into grey matter, white matter, and cerebrospinal fluid according to tissue probability maps. The deformation fields acquired from this process were

used for normalization of functional images into standard Montreal Neurological Institute (MNI) space, and during normalization the functional images were resampled into 2 mm isotropic voxel size. After normalization, the functional images were spatially smoothed by a Gaussian kernel (8 mm, full-width at half-maximum). All of the steps after Topup correction were performed via the SPM12 toolbox (The Wellcome Department of Cognitive Neurology, University College London, London, UK).

**General linear modeling of functional images.** We performed whole-brain analyses of functional images using GLM. For GLM of encoding runs, we applied a boxcar function modeled at the beginning of the reward cue presentation extending to the end of scene encoding, resulting in 3.5 s in total, for our main regressor of interest (Table 1). Regarding GLM for retrieval runs, an impulse function was modeled at the beginning of the scene stimulus presentation. For regressors of no interest for both encoding and retrieval runs, we included impulse functions modeled to participants' response of no interest (encoding runs: scene classification response; retrieval runs: sure/guess response), missed trials, first-order temporal derivatives for each impulse function, session constants and six motion regressors for every GLM. The rationale for employing a boxcar function for only encoding runs is to assume the cue and encoding phase as a single process.

For the group-level analyses of encoding runs, beta coefficients for 16 kinds of regressors of interest, which are the product of 4 (TD: 0, 1, 7, 28 days) x 2 (subsequently remembered/forgotten) x 2 (retrieved 15 minutes/7 days later) conditions, were forwarded to a group analyses where we performed analysis of variance (ANOVA). We tested eight comparisons: (1) main effect of subsequent memory (remembered vs. forgotten) within items retrieved recently, (2) main effect of subsequent memory within items retrieved remotely, (3) main effect of value (0 days > 1 day > 7 days > 28 days), (4) main effect of TD (28 days > 7 days > 1 day > 0 days), (5) 4 x 2 interaction of the subjective value of delayed reward (0, 1, 7, 28 days) and subsequent memory (remembered vs. forgotten) within recent retrieval, (6) 4 x 2 interaction of the subjective value of delayed reward (0 > 1 > 7 > 28 days) and subsequent memory (remembered vs. forgotten) within remote retrieval (7) 4 x 2 interaction of TD of reward (28 > 7 > 1 > 0 days) and subsequent memory (remembered vs. forgotten) within recent retrieval, and (8) 4 x 2 interaction of TD of reward (28 > 7 > 1 > 0 days) and subsequent memory (remembered vs. forgotten) within remote retrieval.

For GLM analysis of fMRI data for retrieval runs, we employed four kinds of regressors of interest: old stimuli which were correctly classified as old (i.e., HIT), old stimuli which were incorrectly classified as new (i.e., MISS), correct rejection of new stimuli, and junk regressors which were comprised of new stimuli incorrectly classified as old (false alarms) and no responses. For each 'old' regressors (HIT and MISS conditions), two parametric modulators were added according to the delayed reward it was associated during the encoding run: subjective value and objective TD. For subjective value, aforementioned parametric modulators derived by multiplying each trial's object reward value by each delay's discounted ratio of each participant (18.9 KRW $^*$ DV of each delay) were entered. On the other hand, the parametric modulators for the objective delay are $10^{-5}$, 1, 7, and 28 for 0 days, 1 day, 7 days, and 28 days, respectively. Since more than one parametric modulator was entered for one onset, the 'order effect' of entering multiple parametric modulators specific to SPM arises (see [33] for more details). Therefore, we performed two GLMs for each parametric regressor of interest by switching the order of parametric regressors entered to acquire statistics for each modulator, respectively (i.e., 1st GLM: subjective value as the first modulator, objective distance as the second modulator; 2nd GLM: vice versa). Finally, the statistical parametric map for parametrically modulated HIT of each participant acquired from 1st level analysis entered 2nd level analyses which performed t-tests against 0. We assumed that the significant voxels would reflect

**Table 1. Summary of models used for fMRI analyses.**

| Model name | Method | Regressors of interest | Contrasts | Contrast vectors |
|---|---|---|---|---|
| Encoding | Factorial design (ANOVA) | 1) Day0_R_Ret0<br>2) Day1_R_Ret0<br>3) Day7_R_Ret0<br>4) Day28_R_Ret0<br>5) Day0_F_Ret0<br>6) Day1_F_Ret0<br>7) Day7_F_Ret0<br>8) Day28_F_Ret0<br>9) Day0_R_Ret7<br>10) Day1_R_Ret7<br>11) Day7_R_Ret7<br>12)Day28_R_Ret7<br>13) Day0_F_Ret7<br>14) Day1_F_Ret7<br>15) Day7_F_Ret7<br>16) Day28_F_Ret7 | (1) Main effect of subsequent memory for recent retrieval<br>(2) Main effect of subsequent memory for remote retrieval<br>(3) Main effect of value<br>(4) Main effect of TD<br>(5) Interaction of subjective value of delayed reward and subsequent memory within recent retrieval<br>(6) Interaction of subjective value of delayed reward and subsequent memory within remote retrieval<br>(7) Interaction of TD of reward and subsequent memory within recent retrieval<br>(8) Interaction of TD of reward and subsequent memory within remote retrieval | (1) [1 1 1 1 –1 –1 –1 –1 0 0 0 0 0 0 0 0]<br>(2) [0 0 0 0 0 0 0 0 1 1 1 1 –1 –1 –1 –1]<br>(3) [3 1 –1 –3 3 1 –1 –3 3 1 –1 –3 3 1 –1 –3]<br>(4) [-3 –1 1 3 –3 –1 1 3 –3 –1 1 3 –3 –1 1 3]<br>(5) [3 1 –1 –3 –3 –1 1 3 0 0 0 0 0 0 0 0]<br>(6) [0 0 0 0 0 0 0 0 3 1 –1 –3 –3 –1 1 3]<br>(7) [-3 –1 1 3 3 1 –1 –3 0 0 0 0 0 0 0 0]<br>(8) [0 0 0 0 0 0 0 0 –3 –1 1 3 3 1 –1 –3] |
| Recent retrieval: PM of value | Parametric modulation (t-test against 0) | 1) HIT_main<br>2) HIT_pm_value<br>3) HIT_pm_TD<br>4) MISS_main<br>5)MISS_pm_value<br>6) MISS_pm_TD<br>7)Correct rejection<br>8) Junk | Parametric effect of value on HIT | [0 1 0 0 0 0 0 0] |
| Recent retrieval: PM of TD | Parametric modulation (t-test against 0) | 1) HIT_main<br>2) HIT_pm_TD<br>3) HIT_pm_value<br>4) MISS_main<br>5) MISS_pm_TD<br>6)MISS_pm_value<br>7)Correct rejection<br>8) Junk | Parametric effect of TD on HIT | [0 1 0 0 0 0 0 0] |
| Remote retrieval: PM of value | Parametric modulation (t-test against 0) | 1) HIT_main<br>2) HIT_pm_value<br>3) HIT_pm_TD<br>4) MISS_main<br>5)MISS_pm_value<br>6) MISS_pm_TD<br>7)Correct rejection<br>8) Junk | Parametric effect of value on HIT | [0 1 0 0 0 0 0 0] |
| Remote retrieval: PM of TD | Parametric modulation (t-test against 0) | 1) HIT_main<br>2) HIT_pm_TD<br>3) HIT_pm_value<br>4) MISS_main<br>5) MISS_pm_TD<br>6)MISS_pm_value<br>7)Correct rejection<br>8) Junk | Parametric effect of TD on HIT | [0 1 0 0 0 0 0 0] |

Abbreviations are like the following. PM = parametric modulation, Day0/1/7/28 = reward delivered after 0/1/7/28 days, Ret0 = retrieved 15 minutes later, Ret7 = retrieved 1 week later, R/F = subsequently remembered/forgotten.

parametric neural activation according to each condition (subjective value or objective delay) during memory retrieval.

In addition to GLM, we performed small-volume correction (SVC) based on a priori regions of interest (ROI) using Harvard-Oxford atlas (Center for Morphometric Analysis). For encoding runs, we selected regions likely to be involved in reward-facilitated memory

consolidation such as the HC [34] or scene-related memory processing such as the parahippocampal cortex [35]. We checked whether the striatum and the orbitofrontal cortex (OFC), regions known to be involved in value processing of monetary reward [10], are involved in representing the main effect of subjective value. Furthermore, for retrieval runs, we selected previously defined regions of the 'retrieval success network' such as frontoparietal control regions and the caudate as ROIs for HIT vs. correct rejection contrast [36].

The multiple comparisons issue upon reporting significant voxels from GLM was dealt with by applying the 3dClustSim function in AFNI (National Institute of Mental Health, Bethesda), in which we entered the median smoothness values of each participant's residual maps and the grey matter mask as input. As a result, the function yielded a set of cluster thresholds that correspond to the alpha value of $p < 0.05$ in a one-tailed test and 18-connected cluster connectivity (edges and faces touch; NN = 2), following the default setting of SPM [37]. Entering residual maps from 22 subjects resulted in 102 contiguous voxels upon $p$-threshold of 0.001 or 2896 contiguous voxels under $p < 0.05$. The results of the following analyses follow the latter criterion for reaching the alpha value of $p < 0.05$.

**Psychophysiological interaction.** The previous findings that the interaction between the HC and the neocortex during memory formation predicts successful encoding [38] and a recent study which found bidirectional informational flow between the neocortex and the HC during memory encoding and retrieval [39] led us to test the possibility of temporal-cue processing regions relaying their information to the HC during successful encoding. To do so, we performed a psychophysiological interaction (PPI) analysis on remotely retrieved trials, with the HC as a seed region. To acquire the variables needed for PPI, we performed an additional GLM on the encoding runs and the regressors of interest were 'trials remembered upon remote retrieval modulated by TD (R-pmod)' and 'trials forgotten upon remote retrieval modulated by TD (F-pmod)'. We used R-pmod labeled as 1 and F-pmod labeled as -1 for the psychological variable, and eigenvariate time series of HC pertaining to R-pmod and F-pmod as a physiological variable. The interaction term was obtained by first deconvolving the physiological variable with the hemodynamic response function (HRF) to acquire the assumed neural response, then multiplying the neural response by psychological variables (R-pmod as 1 and F-pmod as -1), and convolving it with the HRF. These three variables (psychological variable, physiological variable and the interaction term) of interest and variables of no interest such as motion regressors and block effects were put into the GLM. The beta values pertaining to the interaction term in the GLM were entered into a group analysis.

## Results

### Behavioral results

First, we ensured that the task-related performance during the encoding task was independent of possible differences in memory performance among conditions by performing one-way ANOVA on RT and classification accuracy. First, RT among conditions did not show significant differences (mean ± standard deviation (SD); 0 days: 424.87 ± 30.22 ms, 1 day: 427.35 ± 33.67 ms, 7 days: 427.57 ± 31.95 ms, 28 days: 429.9 ± 27.61 ms). Second, the mean classification accuracy for each condition showed a ceiling effect (0 days: 93.46 ± 5.75%, 1 day: 92.63 ± 5.84%, 7 days: 92.01 ± 5.27%, 28 day: 93.25 ± 6.12%) as well as no significant differences among conditions. Therefore, in further analyses, we claim that the task-related performance of each condition during the encoding session does not account for possible differences in memory performance of each condition.

Participants' CR was assessed by subtracting hit rate (the number of hits divided by total possible hits in the given condition) by false-alarm rate (the number of false alarms divided by

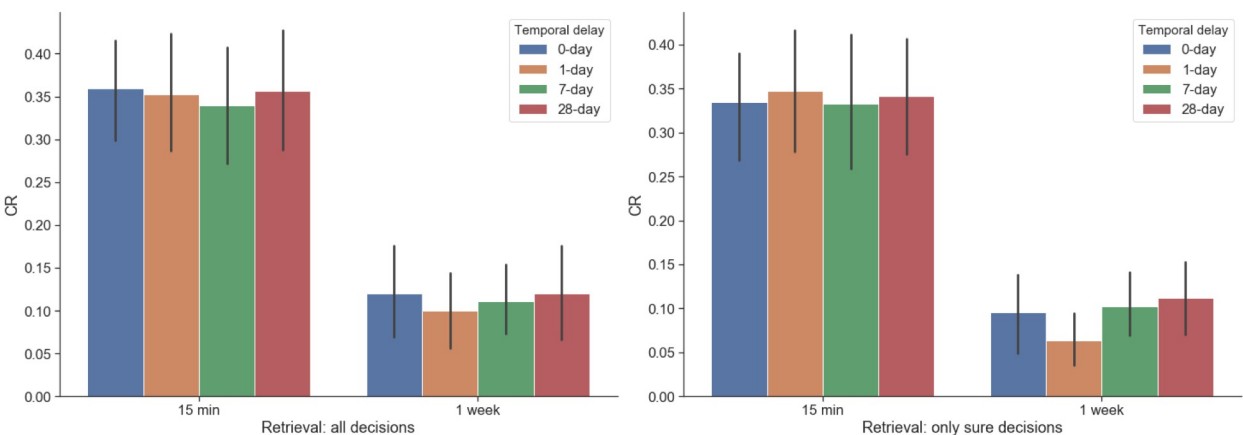

**Fig 2. Corrected recognition (CR).** Error bars indicate standard deviation, 15 min = retrieval right after encoding, 1 week = retrieval 1 week after encoding, (left) CR using both sure and guessed responses (right) CR using only sure responses.

total possible false alarms, identical across all conditions), regardless of the confidence rating for each decision (Fig 2). First, participants' CR was subjected to a 2 x 4 ANOVA (recent vs. remote retrieval x 4 different delays of reward). Only the main effect of retrieval time point turned out to be significant ($F(1,21) = 122.4$, $p < 0.001$). Post hoc analysis of retrieval revealed that CR was significantly higher for items that were retrieved 15 minutes later in contrast to items retrieved one week later ($t(22) = 8.64$, $p < 0.001$), as expected. Although the main-effect of reward-delay conditions turned out to be insignificant in ANOVA using both sure and guess decisions as well as ANOVA using only sure decisions, we performed an exploratory t-test using only sure decisions. Here, we performed a paired t-test after concatenating 0 days and 1 day's sure HITs as 'sooner' and 7 days and 28 days as 'later' conditions in order to make a straightforward interpretation. This discretization has been performed on previous temporal discounting studies [40], which is justified by the fact that participants' choice behavior, which can be converted to subjective value, is linearly related to temporal delay. This exploratory t-test revealed that the 'later' condition's CR is marginally higher than the 'sooner' condition CR ($t(22) = 2.06$, $p = 0.05$).

Participants' discount rate ($k$) ranged from 0.0002 to 0.1112 (mean ± standard deviation (SD); 0.0209 ± 0.0309). When assuming subjective value for 0-day as 18.9 KRW, the subjective value for other days were all significantly lower than 0-day (1 day = 18.53 ± 0.53 KRW, $t(21) = 3.28$, $p = 0.004$; 7-days = 16.92 ± 2.41 KRW, $t(21) = 3.86$, $p = 0.001$; 28-days = 14.06 ± 4.38 KRW, $t(21) = 5.19$, $p < 0.001$).

### Neuroimaging data

**Replication of previous literature.** We performed a set of analyses to check whether results from previous studies can be replicated and to strengthen the results from the main analyses. In encoding runs, SVC yielded activation for the parahippocampal cortex pertaining to subsequently remembered vs. forgotten items retrieved one week later (MNI coordinates x, y, z = -34, -6, -22; $pSVC = 0.003$; $k = 25$; $z = 2.75$), and activation of the striatum (-24, -10, 8; $pSVC = 0.009$; $k = 20$; $z = 2.37$) and the OFC (32, 30, -22; $pSVC = 0.012$; $k = 42$; $z = 2.27$) when checking for regions showing a main effect of subjective value. For the first retrieval session, the caudate (8, 8, -4; $pSVC < 0.001$; $k = 27$; $z = 3.79$) was activated for retrieval success upon SVC. The same contrast applied to the second retrieval session which yielded whole-brain level activation in a cluster comprising the right inferior parietal lobule, precuneus, posterior

cingulate, and a cluster of regions in the frontal lobe (Table 2). All these regions have been reported to be activated upon direct HIT vs. correct rejection contrasts [36]. SVC in addition to whole-brain analysis returned activation in the bilateral caudate (left: -12, 8, 8; $pSVC < 0.001$; $k = 165$; $z = 3.64$; right: 12, 6, 10; $pSVC = 0.001$; $k = 201$; $z = 3$).

**Main analyses.** Among eight contrasts within encoding runs, only three contrasts yielded significant activations at the whole-brain level: (1) main effect of TD, (2) interaction of TD of reward ($28 > 7 > 1 > 0$ days) and subsequent memory (remembered vs. forgotten) within remote retrieval, and (3) interaction of TD of reward and subsequent memory within recent retrieval (Table 3). First, a cluster of regions encompassing the precentral gyrus, postcentral gyrus, superior temporal gyrus, insula, middle temporal gyrus, superior temporal gyrus, inferior frontal gyrus, cingulate gyrus, and the medial frontal gyrus showed activation in correlation with the TD of the reward, regardless of other conditions (Fig 3). Second, a cluster including the precentral gyrus, cerebellum, middle temporal gyrus, cingulate gyrus, caudate, anterior cingulate, inferior temporal gyrus, inferior occipital gyrus, precuneus, paracentral lobule, and the fusiform gyrus was activated upon searching for regions showing an SME proportionate to TD for items retrieved recently (Fig 4). Third, a cluster encompassing the frontal lobe, insula, and the precentral gyrus, and a cluster comprised of the temporal gyrus, frontal lobe, and the inferior parietal lobule showed an SME proportionate to the more distant TD for items retrieved remotely (Fig 5).

Next, for analyses of retrieval runs, we examined areas showing proportionate activation to formerly associated value or TD of the retrieved stimuli. Significant whole-brain level activation was only found for TD during remote retrieval: a range of areas encompassing the bilateral superior frontal gyrus, middle frontal gyrus (MFG), and the medial frontal gyrus (Fig 6 and Table 4).

Since it was unexpected that the PFC represents the TD of reward associated with the stimuli during both encoding and remote retrieval, we performed a conjunction analysis to further clarify which regions are activated for both encoding and retrieval. Upon the threshold of $p < 0.001$ with at least ten contiguous voxels for conjunction, the MFG, insula, superior frontal gyrus, precentral gyrus, superior temporal gyrus, cerebellum, inferior frontal gyrus, and the medial frontal gyrus were activated (Table 5 and Fig 7). We performed a post hoc analysis to examine the SME of the MFG, the peak showing the greatest activation among regions

**Table 2. Replication: Activation peaks for HIT vs. correct rejection within remote retrieval runs.**

| Contrast | Cluster | Voxels | BA | Hemisphere | Region name | z Stat | X | Y | Z |
|----------|---------|--------|-----|------------|-------------|--------|-----|-----|-----|
| **HITvsCR** | 1 | 4339 | 7 | Right | Precuneus | 3.93 | 14 | -76 | 50 |
| | | | 31 | Right | Posterior Cingulate | 3.00 | 22 | -58 | 22 |
| | | | 7 | Left | Precuneus | 2.94 | -18 | -72 | 44 |
| | | | 40 | Right | Inferior Parietal Lobule | 2.56 | 52 | -40 | 50 |
| | | | 19 | Right | Middle Occipital Gyrus | 2.48 | 40 | -88 | 20 |
| | | | 7 | Right | Superior Parietal Lobule | 2.24 | 32 | -60 | 54 |
| | | | 19 | Right | Cuneus | 2.23 | 30 | -92 | 28 |
| | 2 | 3197 | 10 | Left | Inferior Frontal Gyrus | 3.73 | -48 | 42 | -2 |
| | | | 8 | Right | Middle Frontal Gyrus | 3.60 | 44 | 38 | 44 |
| | | | 9 | Right | Superior Frontal Gyrus | 3.59 | 46 | 48 | 38 |
| | | | 46 | Left | Middle Frontal Gyrus | 2.98 | -44 | 54 | 8 |
| | | | 10 | Left | Superior Frontal Gyrus | 2.65 | -24 | 60 | -8 |

Abbreviation: BA = Brodmann area number.

**Table 3. Whole-brain activation of regions during encoding phase.**

| Contrast | Cluster | Voxels | BA | Hemisphere | Region name | z Stat | X | Y | Z |
|---|---|---|---|---|---|---|---|---|---|
| **1. MAIN: TD** | 1 | 3831 | 43 | Right | Postcentral Gyrus | 3.80 | 58 | -18 | 18 |
| | | | 38 | Right | Superior Temporal Gyrus | 3.76 | 48 | 2 | -12 |
| | | | 13 | Right | Insula | 3.5 | 46 | -36 | 24 |
| | | | 6 | Right | Precentral Gyrus | 3.01 | 52 | -8 | 56 |
| | | | 21 | Right | Middle Temporal Gyrus | 2.27 | 60 | 8 | -14 |
| | 2 | 3752 | 22 | Left | Superior Temporal Gyrus | 3.68 | -38 | -54 | 18 |
| | | | 7 | Left | Precuneus | 2.72 | -8 | -54 | 46 |
| | | | 7 | Right | Precuneus | 2.38 | 2 | -54 | 62 |
| | | | 39 | Left | Middle Temporal Gyrus | 2.23 | -42 | -70 | 24 |
| | 3 | 5144 | 40 | Left | Postcentral Gyrus | 3.38 | -60 | -24 | 16 |
| | | | 6 | Left | Precentral Gyrus | 3.1 | -50 | -2 | 12 |
| | | | 13 | Left | Insula | 2.97 | -36 | 4 | 8 |
| | | | 47 | Left | Inferior Frontal Gyrus | 2.9 | -26 | 12 | -20 |
| | | | 31 | Left | Cingulate Gyrus | 2.8 | -18 | -28 | 42 |
| | | | 6 | Left | Medial Frontal Gyrus | 2.69 | -12 | -10 | 52 |
| **2. IE: SME proportionate to TD when retrieved recently** | 1 | 22436 | 6 | Right | Precentral Gyrus | 3.84 | 32 | 0 | 38 |
| | | | * | Right | Cerebellum | 3.83 | 44 | -58 | -26 |
| | | | 21 | Right | Middle Temporal Gyrus | 3.71 | 48 | -46 | 4 |
| | | | 24 | Left | Cingulate Gyrus | 3.70 | -14 | -4 | 42 |
| | | | * | Left | Caudate | 3.51 | -20 | 26 | 2 |
| | | | 8 | Left | Middle Frontal Gyrus | 3.47 | -32 | 34 | 40 |
| | | | 19 | Right | Inferior Temporal Gyrus | 3.43 | 44 | -60 | -6 |
| | | | 19 | Left | Inferior Occipital Gyrus | 3.39 | -46 | -80 | -6 |
| | | | 24 | Left | Anterior Cingulate | 3.32 | -8 | 24 | 12 |
| | | | 7 | Right | Precuneus | 3.28 | 12 | -50 | 52 |
| | | | * | Left | Cerebellum | 3.25 | -34 | -62 | -10 |
| | | | 5 | Right | Paracentral Lobule | 3.22 | 22 | -44 | 48 |
| | | | 37 | Right | Fusiform Gyrus | 3.16 | 42 | -40 | -8 |
| **3. IE: SME proportionate to TD when retrieved remotely** | 1 | 4434 | 47 | Left | Middle Frontal Gyrus | 4.52 | -42 | 32 | 0 |
| | | | 13 | Left | Insula | 3.73 | -44 | 2 | 12 |
| | | | 8 | Left | Superior Frontal Gyrus | 3.59 | -28 | 26 | 50 |
| | | | 9 | Left | Precentral Gyrus | 3.32 | -34 | 14 | 36 |
| | | | 47 | Left | Inferior Frontal Gyrus | 3.22 | -24 | 24 | -6 |
| | | | 32 | Left | Anterior Cingulate | 3.15 | -8 | 34 | 24 |
| | | | 6 | Right | Superior Frontal Gyrus | 2.47 | 6 | 22 | 58 |
| | 2 | 3387 | 22 | Right | Superior Temporal Gyrus | 3.49 | 68 | -40 | 12 |
| | | | 47 | Right | Inferior Frontal Gyrus | 3.45 | 46 | 36 | -10 |
| | | | 22 | Right | Middle Temporal Gyrus | 3.39 | 52 | -36 | 0 |
| | | | 11 | Right | Middle Frontal Gyrus | 3.12 | 40 | 46 | -10 |
| | | | 13 | Right | Insula | 2.46 | 38 | 20 | 0 |
| | | | 40 | Right | Inferior Parietal Lobule | 2.36 | 52 | -38 | 28 |

Abbreviations: BA = Brodmann area number, MAIN = main effect, IE = interaction effect.

identified by the conjunction analysis. The SME was defined as beta values of 'remembered' subtracted by 'forgotten' in each condition (TD). Pairwise t-tests revealed that activation related to the SME for reward without delay (0 days) was significantly lower than activation at 1 day ($t(22) = 2.234$, $p = 0.037$), 7 days ($t(22) = 3.306$, $p = 0.003$), and 28 days ($t(22) = 4.75$,

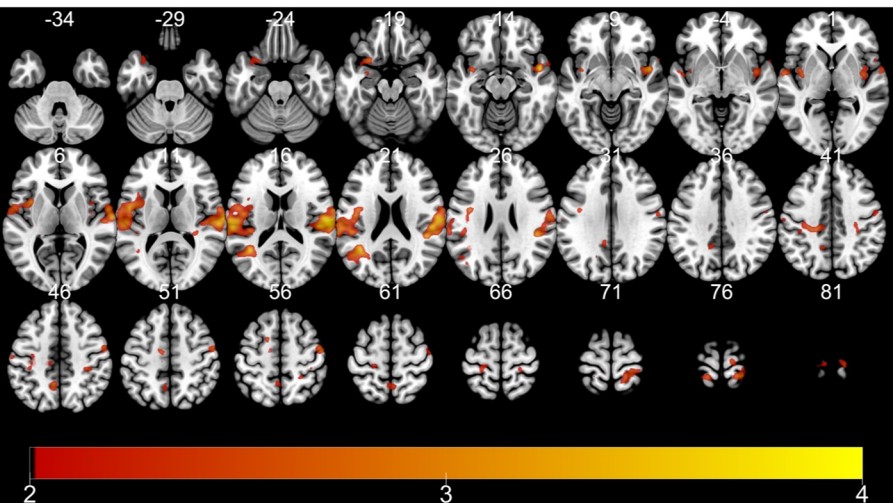

**Fig 3. fMRI results (encoding): (1) Main effect of TD.** Color bar indicates t-statistics.

$p < 0.001$). Furthermore, there was a trend that activation at 1 day is smaller than at 7 days ($t$(22) = 1.77, $p = 0.09$) and 28 days ($t$(22) = 2.48, $p = 0.082$). Therefore, the observed interaction of the SME between conditions was mainly driven by the significantly lower SME of the 0-days condition than by the other conditions.

These observations lead to two intermediate conclusions: (1) the greater the distance of the reward associated with a stimulus, the higher the activation of the MFG both during encoding and remote retrieval, and (2) this role of MFG becomes significant once consolidation takes place, given that this pattern of MFG does not emerge when items are retrieved before consolidation.

**Psychophysiological interaction.** The HC seed used for the eigenvariate time series extraction was a 6 mm sphere around peak coordinate -22, -20, -20 obtained from R-pmod vs. F-pmod contrast in the whole-brain GLM. According to PPI results, regions including MFG and bilateral IPL showed functional connectivity with the HC (Table 6 & Fig 8).

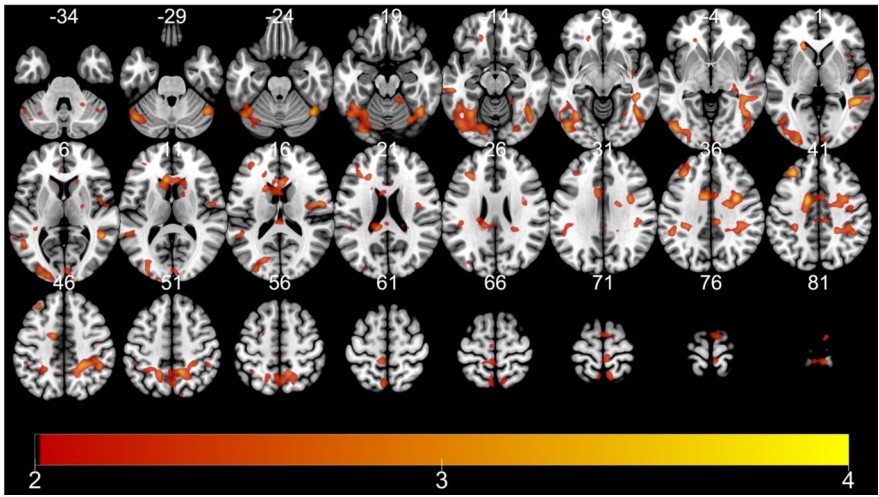

**Fig 4. fMRI results (encoding).** IE: SME proportionate to TD when retrieved recently. Color bar indicates t-statistics.

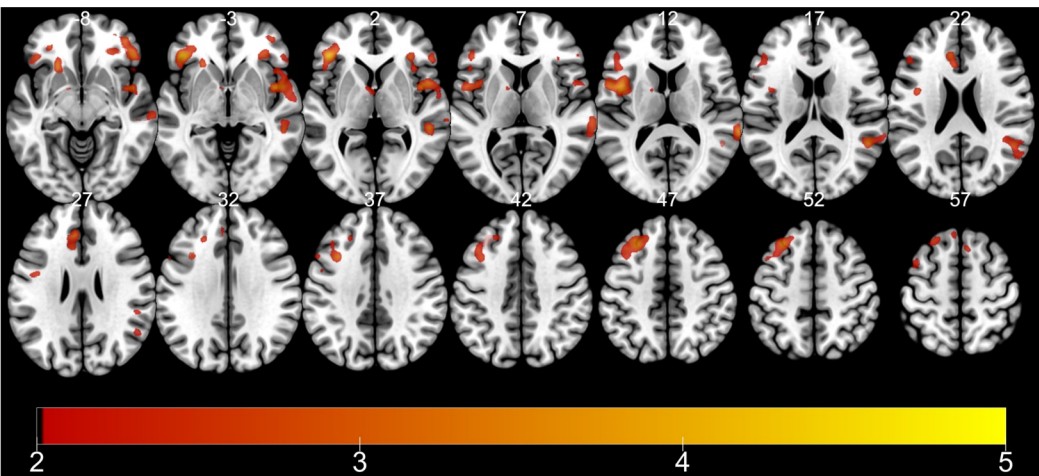

**Fig 5. fMRI results (encoding).** IE: SME proportionate to TD when retrieved remotely. Color bar indicates t-statistics.

## Discussion

Reward-motivated memory encoding and temporal discounting of a delayed reward are two well-studied fields in cognitive neuroscience. But to our knowledge, no previous study has directly linked these domains. This study is the first to examine whether rewards delivered at different TD lead to differential memory effects. We found that temporally distant rewards might lead to better memory performance, which is in contrast to previous studies. We showed that a) when considering only confident retrieval (sure decisions), behavioral measure of memory performance (CR) showed a trend that the more reward is delayed, the higher is the chance of memory consolidation, b) in the encoding session, while the subjective value of the reward did not yield activation proportional to an SME as expected, TD did, and c) some of these regions were again observed during the second retrieval session when HIT activation was parametrically modulated by TD associated with the stimuli.

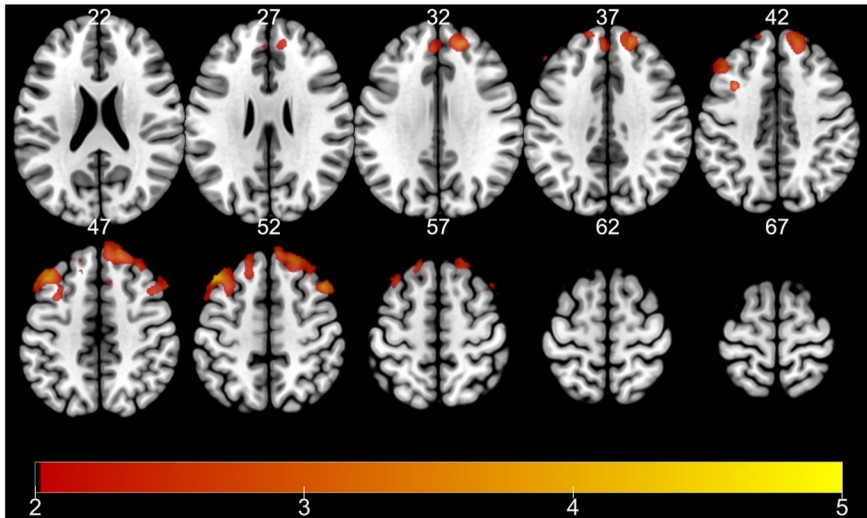

**Fig 6. fMRI results: HITs showing commensurate activation with TD during remote retrieval.** Color bar indicates t-statistics.

**Table 4. Activation peaks for regions showing proportionate SME for TD for items retrieved 1 week later.**

| Cluster | Voxels | BA | Hemisphere | Region name | z Stat | X | Y | Z |
|---|---|---|---|---|---|---|---|---|
| 1 | 3732 | 8 | Left | Superior frontal gyrus | 4.04 | -44 | 26 | 52 |
| | | 8 | Right | Superior frontal gyrus | 3.57 | 46 | 22 | 54 |
| | | 6 | Left | Middle frontal gyrus | 3.12 | -34 | 12 | 42 |
| | | 6 | Left | Medial frontal gyrus | 2.57 | -4 | 44 | 36 |

Abbreviation: BA = Brodmann area number.

Our result seems to be in conflict with the preexisting literature on reward-motivated memory encoding and temporal discounting. While previous studies showed that a) reward is devalued as its delivery is delayed [14] and b) reward magnitude is positively correlated with the SME [10], our result suggests that a delayed reward acts as if it is more valued (when we put the emphasis on b) or—although reward is temporally discounted—a reward associated with TD might somehow involve a different mechanism than dopaminergic reward-modulation (when we put the emphasis on a). Since we have collected data from a delay-discounting decision-making task and found that participants showed temporal discounting behavior as reported in previous studies, it seems more plausible to assume the latter hypothesis that reward in a temporal context cannot be solely interpreted in light of previous reward-motivated memory literature. Thus, we seek a novel account to explain this unprecedented SME driven by temporally distant rewards.

To do so, we start by discussing the main effect of TD of reward during encoding that perceiving temporally distant reward engages lateral prefrontal and parietal activation (4th contrast of Encoding GLM; see Table 1). In fact, distant reward cues leading to lateral PFC or parietal activation are not unfamiliar. Lateral prefrontal activation has been reported after choosing delayed reward vs. immediate reward in intertemporal decision-making tasks [41,42]. However, linking these studies with the current results requires caution because intertemporal decision-making tasks typically involve situations in which participants are presented with larger-delayed and smaller-immediate rewards, thereby provoking self-control to inhibit impulsivity when participants choose the former over the latter, especially when taking into account that frontoparietal regions are representative members of the control network [43]. If lateral prefrontal activity reported in these studies is solely the product of executive control

**Table 5. Activation peaks for regions showing proportionate activation for TD during both encoding and remote retrieval ($p < .001$).**

| Voxels | BA | Hemisphere | Region name | z Stat | X | Y | Z |
|---|---|---|---|---|---|---|---|
| 88 | 47 | Left | Middle Frontal Gyrus | 4.52 | -42 | 32 | 0 |
| 57 | 13 | Left | Insula | 3.73 | -44 | 2 | 12 |
| 34 | 8 | Left | Superior Frontal Gyrus | 3.59 | -28 | 26 | 50 |
| 36 | 9 | Right | Precentral Gyrus | 3.54 | 36 | 14 | 38 |
| 14 | 22 | Right | Superior Temporal Gyrus | 3.49 | 68 | -40 | 12 |
| 12 | * | Right | Cerebellum | 3.49 | 12 | -28 | -6 |
| 17 | 47 | Right | Inferior Frontal Gyrus | 3.45 | 46 | 36 | -10 |
| 44 | 22 | Right | Superior Temporal Gyrus | 3.43 | 50 | 6 | 0 |
| 13 | 10 | Left | Medial Frontal Gyrus | 3.38 | -14 | 52 | 0 |
| 27 | 9 | Right | Medial Frontal Gyrus | 3.33 | 16 | 44 | 20 |

Abbreviation: BA = Brodmann area number.

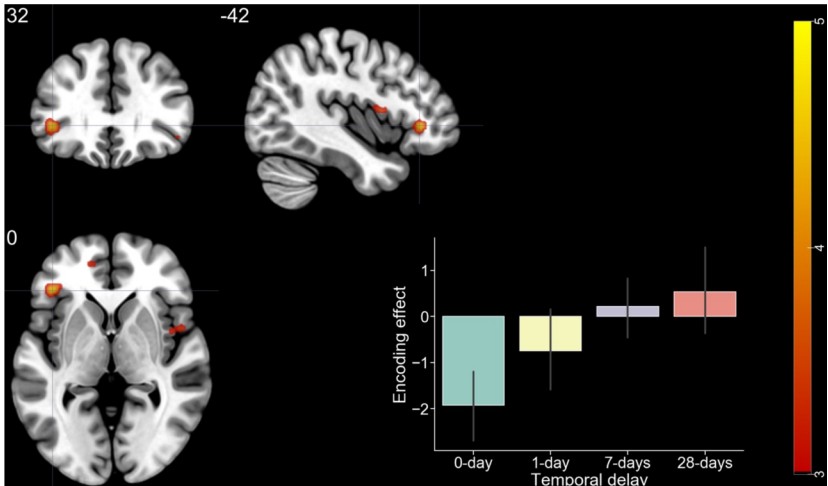

**Fig 7. fMRI results: Encoding-retrieval overlap.** Color bar indicates t-statistics, bar plot indicates SME (beta values of Remembered subtracted by Forgotten) of medial frontal gyrus (-42, 32, 0). Error bar indicates standard deviation.

rather than TD, it would not support our results since our study does not require choosing between delayed and immediate rewards.

However, a study which examined the prediction of future rewards outside the intertemporal decision-making paradigm also reported that in the long vs. short delay contrast, the ventrolateral PFC (vlPFC), insula, dorsolateral PFC (dlPFC), dorsal premotor cortex, and the inferior parietal cortex are activated [44]. In addition, a study that examined single-neuron activity of rats revealed neurons in the OFC/ventromedial PFC (vmPFC) which fire more

**Table 6. Activation peaks of PPI results.**

| Voxels | BA | Hemisphere | Region name | z Stat | X | Y | Z |
|---|---|---|---|---|---|---|---|
| **8373** | 6 | Left | Middle Frontal Gyrus | 4.01 | -26 | -12 | 44 |
| | 40 | Left | Inferior Parietal Lobule | 3.25 | -64 | -38 | 40 |
| | 31 | Left | Cingulate Gyrus | 3.21 | -18 | -40 | 40 |
| | 29 | Right | Posterior Cingulate | 3.15 | 12 | -42 | 12 |
| | 46 | Left | Inferior Frontal Gyrus | 3.01 | -50 | 36 | 8 |
| | 43 | Left | Postcentral Gyrus | 2.84 | -54 | -8 | 14 |
| | 13 | Left | Insula | 2.82 | -32 | 22 | 14 |
| | 2 | Left | Postcentral Gyrus | 2.79 | -68 | -24 | 30 |
| | 5 | Left | Paracentral Lobule | 2.78 | -16 | -30 | 46 |
| | 7 | Right | Precuneus | 2.74 | 24 | -60 | 52 |
| | 6 | Left | Precentral Gyrus | 2.69 | -68 | 0 | 16 |
| | 6 | Left | Medial Frontal Gyrus | 2.66 | -14 | -22 | 54 |
| **4954** | * | Right | Caudate | 3.84 | 26 | -34 | 8 |
| | 37 | Right | Fusiform Gyrus | 3.35 | 36 | -42 | -10 |
| | 2 | Right | Postcentral Gyrus | 3.25 | 60 | -22 | 56 |
| | 40 | Right | Inferior Parietal Lobule | 2.59 | 68 | -30 | 32 |
| | 40 | Right | Inferior Parietal Lobule | 2.52 | 48 | -28 | 30 |
| | 13 | Right | Insula | 2.41 | 42 | -20 | 20 |

Abbreviation: BA = Brodmann area number.

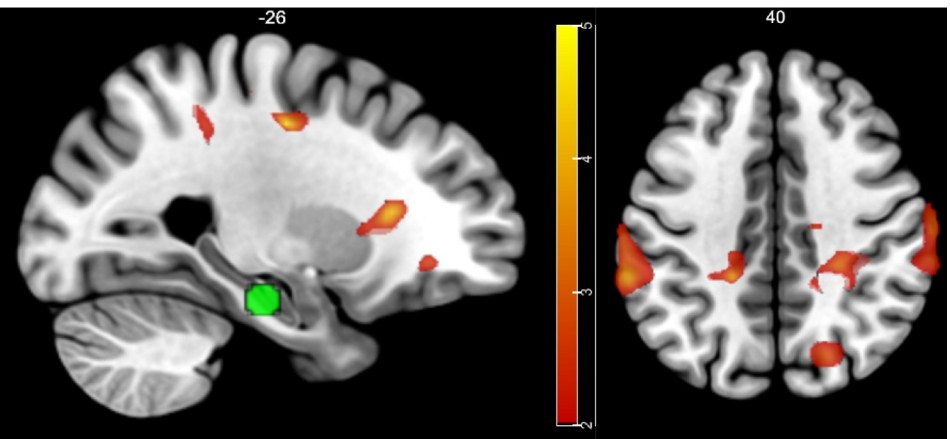

**Fig 8. fMRI results: PPI.** Color bar indicates t-statistics. (Left) The seed region, the HC (-22, -20, -20), is represented as a green sphere.

strongly for delayed than for immediate rewards [45]. Frontoparietal regions have been reported to be activated for processing time and future perception in general and it has been previously reported that elaborating a future event activates the inferior frontal gyrus and the supramarginal gyrus [46]. Additionally, studies on macaques have found activations in the dlPFC and the inferior parietal cortex [47] and the posterior parietal cortex [48] during time perception tasks. These comprehensive studies weaken the possibility that frontoparietal activation associated with choosing a distant reward in intertemporal decision-making tasks solely reflects self-control. Instead, they may indicate that frontoparietal regions also process some portion of temporal information upon perceiving a delayed reward.

Interestingly, in our study, some of these regions such as the PFC or the inferior parietal lobule (IPL) go further from merely representing the TD of reward (4th contrast of Encoding GLM) to showing an SME commensurate with the TD during memory encoding (8th contrast of Encoding GLM). This is supported by the PPI result that PFC and parietal regions are likely to be transferring their temporal information to the HC during encoding, though causality is not defined in our study. This is consistent with a previous study reporting that the PFC and the IPL are also members of the retrieval success network [36] as the PFC and the IPL are activated upon successful retrievals in contrast to correct rejections in memory retrieval tasks. While it appears to be counterintuitive to find the answer to the 'encoding' results from the 'retrieval' phase, the principle of transfer-appropriate processing supports this hypothesis [49,50]. It assumes that memory will be enhanced if memory retrieval reactivates similar regions as those that were engaged during cognitive operations of encoding. Cortical reinstatement, which refers to the reactivation of the original neural trace of a stimulus during retrieval [51], might be an example, since the degree of cortical reinstatement has been found to correlate with memory recall performance [52,53]. It has also been found that perception-retrieval overlap of the perirhinal cortex and the parahippocampal cortex predicts success recall for object- and scene-stimuli, respectively [54]. Given this background, it is not surprising that brain regions that respond to TD during encoding, which are also key regions contributing to retrieval success, enhance memory for items associated with distant reward by the coincidental encoding-retrieval overlap.

However, even if we assume that the SME of delayed reward is proportionate to the TD due to encoding-retrieval overlap of retrieval success (or 'temporal-distance') regions, it is conditional because this phenomenon is only observed when memory is retrieved one week but not

15 minutes later. This is an important argument because we are trying to provide an alternative account to the dopaminergic account, but the differential effect of reward on memory observed pre- and post-consolidation is what would be expected by the dopamine account. To resolve this, we focus on the fact that the remote retrieval took place one week after encoding in our study. Consolidation continues even after 24 hours, reorganizing the initially constructed engram of memory at a system (network) level [55]. One of the accounts regarding system consolidation of memory proposes that while initially formed memory is rich in episodic details, supported by the HC, it eventually becomes abstract and semantic over time, thereby becoming less dependent on the HC [56]. Indeed, an fMRI study over three months has shown that while recently formed memory relies more on hippocampal representations, systems consolidation shifts the site of memory from the HC to neocortical areas such as the mPFC over time [57]. In light of this, our results can be explained by the following: memory performance for recent retrieval is not different among conditions because while recently formed episodic details depend on hippocampal activation, the four temporal conditions did not show differences in HC activation. However, as the HC contributes less to the network, regions which initially represented cognitive information become the site of memory upon encountering the stimulus. In our study, the regions initially coding cognitive information (i.e., TD) such as the PFC showed differential activation for temporal conditions during encoding, and as a function of TD. This initial difference stood out more as memory becomes more dependent on traces in the PFC. This may be one of the reasons that stimuli associated with a more distant reward, triggering stronger activations in the neocortex, became more likely to be remembered as retrieval becomes less dependent on the HC.

This study is the first to examine how the TD of rewards modulates episodic memory and the dynamics of memory retrieval with time. This study is important because it suggested that temporally varied rewards may not act like rewards which vary in magnitude in the brain. However, this study has limitations because the behavioral effects are statistically weak. Furthermore, the results from this study could also be analyzed by pattern analyses such as multivariate pattern analyses so that representations among temporal conditions for encoding and retrieval can be directly compared.

## Supporting information

**S1 Table. Minimal raw behavioral data for all subjects used in this study's analyses.** https://figshare.com/s/c34211f53f14c7bae391. Yoo, Jungsun; Min, Seokyoung; Lee, Seung-Koo; Han, Sanghoon (2020): SupportingInformation1_minimal_data. figshare. Dataset. https://doi.org/10.6084/m9.figshare.13285928.v1.
(XLSX)

## Author Contributions

**Conceptualization:** Jungsun Yoo, Seokyoung Min, Sanghoon Han.

**Data curation:** Jungsun Yoo, Seokyoung Min.

**Formal analysis:** Jungsun Yoo.

**Funding acquisition:** Sanghoon Han.

**Investigation:** Jungsun Yoo, Sanghoon Han.

**Methodology:** Jungsun Yoo, Seokyoung Min, Seung-Koo Lee, Sanghoon Han.

**Project administration:** Jungsun Yoo.

**Resources:** Seung-Koo Lee, Sanghoon Han.

**Supervision:** Seung-Koo Lee, Sanghoon Han.

**Visualization:** Jungsun Yoo.

**Writing – original draft:** Jungsun Yoo.

**Writing – review & editing:** Jungsun Yoo.

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
