## [Decision Letter · Decision Letter 0]

9 Oct 2020

PONE-D-20-24115

Neural correlates of episodic memory modulated by temporally delayed rewards

PLOS ONE

Dear Dr. Han,

Thank you for submitting your manuscript to PLOS ONE. After careful consideration, we feel that it has merit but does not fully meet PLOS ONE’s publication criteria as it currently stands. Therefore, we invite you to submit a revised version of the manuscript that addresses the points raised during the review process.

We look forward to receiving your revised manuscript.

Kind regards,

Qiyong Gong, M.D., Ph.D.,

Academic Editor

PLOS ONE

Journal Requirements:

2. Please change "female” or "male" to "woman” or "man" as appropriate, when used as a noun.

3.  Please provide additional details regarding how you determined the optimal sample size for your study.

4. Please ensure that you refer to Figure 6 in your text as, if accepted, production will need this reference to link the reader to the figure.

5. We note you have included a table to which you do not refer in the text of your manuscript. Please ensure that you refer to Table 3 in your text; if accepted, production will need this reference to link the reader to the Table.

Reviewers' comments:

Reviewer's Responses to Questions

**Comments to the Author**

1. Is the manuscript technically sound, and do the data support the conclusions?

Reviewer #1: Partly

Reviewer #2: Yes

2. Has the statistical analysis been performed appropriately and rigorously? 

Reviewer #1: No

Reviewer #2: Yes

3. Have the authors made all data underlying the findings in their manuscript fully available?

Reviewer #1: No

Reviewer #2: Yes

4. Is the manuscript presented in an intelligible fashion and written in standard English?

Reviewer #1: Yes

Reviewer #2: Yes

5. Review Comments to the Author

Reviewer #1: This study's authors develop a new experimental paradigm to examine the behavioral and neural correlates of episodic memory modulated by temporally delayed rewards. As the authors noted, it is novel to link reward-motivated memory encoding and temporal discounting together. This study's major findings are that the PFC activation is proportional to the degree of temporal discounting during the encoding and retrieval phase of the scene recognition memory task.

In general, the writing of this manuscript is sound, and the data-analyses are comprehensive. I appreciated the authors' effort. However, the study is somewhat inconclusive. In particular, the behavioral results of this study are not clear and difficult to explain. Based on Figure 2 and its corresponding results, I did not see a real consistent data pattern. The authors' explanation on the "later" (7 and 28 days) vs. "sooner" (0 and 1 day) contrast might be posthoc. Since the behavioral results are not clear, I am not sure that the fMRI results really reflect the delayed reward's memory-enhanced effect. Instead, the fMRI results may merely reflect perceptual differences of temporal delays of different cues. Together, the inconclusive nature of data interpretation might lowers the scientific value of this study.

Reviewer #2: The authors utilized both behavioral and fMRI methods to investigate whether more immediate rewards are associated with better long-term memory. However, the findings were contrary to this hypothesis. They explained the unexpected results by the encoding-retrieval process, which is mainly represented by the LPFC activation.

The experiments were well designed and conducted. The statistical analyses were reliable. The whole-manuscript was also well-written. I believe that this is a good study although the authors got findings contrary to their hypotheses.

6. PLOS authors have the option to publish the peer review history of their article (what does this mean?). If published, this will include your full peer review and any attached files.

Reviewer #1: No

Reviewer #2: No

---

## [Author Response · Author response to Decision Letter 0]

2 Dec 2020

Rebuttal Letter

Dear Dr. Gong, 

First of all, we appreciate that our manuscript, Neural correlates of episodic memory modulated by temporally delayed rewards, has been carefully evaluated by the editor and the reviewers and given an opportunity to be revised. The editor’s and reviewers’ suggestions were very helpful, and we have changed our manuscript accordingly in our revised manuscript. We have highlighted the modifications in our revised manuscript by highlighting the modified text in yellow and coloring the text in red in correspondence with the reviewers’ and the editor’s comments, respectively. In the remaining part of this letter, we provide our detailed response to each point raised by the reviewers, which adheres to the following format: a) comments from editors or reviewers are in bold text, and b) our corresponding responses are shown in plain text. The detailed response is followed by a list of references we used to make our claims. We again appreciate that our manuscript was considered to potentially have merit after careful consideration of the editor and the reviewers and would like to thank them for the helpful comments. We sincerely hope that the editor and the reviewers find the revisions satisfactory. 

Kind regards, 

Jungsun Yoo, Seokyoung Min, Seung-Koo Lee, and Sanghoon Han

 

Responses to the editor’s comments: 

1. Please ensure that your manuscript meets PLOS ONE's style requirements, including those for file naming. The PLOS ONE style templates can be found at https://journals.plos.org/plosone/s/file?id=wjVg/PLOSOne_formatting_sample_main_body.pdf and https://journals.plos.org/plosone/s/file?id=ba62/PLOSOne_formatting_sample_title_autauth_affiliations.pdf. 

Thank you for pointing this out, and we apologize for submitting the manuscript that does not meet the requirements in formatting. To comply with PLOS ONE’s title, author, and affiliations formatting guidelines, we made modifications to the first page of the manuscript. Specifically, affiliation footnotes were changed to numbers and the current affiliation (current address) of the first author was added and referred to using the relevant symbols. Also, the corresponding author’s initial was added to the e-mail. 

The following changes were made to follow the manuscript body formatting guidelines. First, the Level 1-3 headings were changed according to the guideline – specifically, the font was adjusted, and the sentence case was adapted. Second, pertaining to the file naming for figures, image files for figures have been converted to “.tiff” and renamed to match the labels within the manuscript. Figure captions and citations have been changed. Table captions and table citations have also been modified to meet the PLOS ONE Table Guidelines. The reference citations were also modified to meet the guidelines, and we deleted some uncited literature from the reference list. We hope these changes make the manuscript fit the style requirements as described in the guidelines. 

2. Please change "female” or "male" to "woman” or "man" as appropriate, when used as a noun.

Thank you for your advice. We changed the term female to women accordingly on line 14 on page 5 (under the section Participants and materials).

3. Please provide additional details regarding how you determined the optimal sample size for your study.

This is an important point in a study, and we appreciate that you raised this issue. The information regarding this point is now added to the manuscript on page 6 lines 5-8 (under the section Participants and materials). The sample size of 22 subjects was determined by following the convention of experimental fMRI studies (studies from top-cited journals published during 2017-2018 have reported sample size of 23-24 (Szucs & Ioannidis, 2020)). Although determining optimal sample size by power calculation is considered as one of the methods of justifying sample size for a study (Mumford, 2012), we cannot use this method at this point because the sample size using this calculation should be determined prior to data collection. However, we believe that the fact that we did not determine a priori sample size does not necessarily undermine the validity of our fMRI results; a study that examined the statistical power of “gold standard” sample size (n=58) and “typical” sample size (n=20) reported that, although the statistical power was low for the typical sample, the significantly activated voxels from this subset were generally true positives and tend to be correlated with the activations of the larger group (Murphy and Garavan, 2004). Also, we believe the fact that our final sample size of 22 subjects falls within the range of recently published fMRI studies on episodic memory encoding and/or retrieval (Xiao et al., 2020, n=20; Friedl-Werner et al., 2020, n=22; Miotto et al., 2020, n=25) could serve as another reference point for the sample size of our study. 

4. Please ensure that you refer to Figure 6 in your text as, if accepted, production will need this reference to link the reader to the figure.

We deeply thank you for pointing this out and apologize for not fixing this in advance. Figure 6 is now referenced together with Table 4 on line 6 of page 23 (under the section Main analyses). 

5. We note you have included a table to which you do not refer in the text of your manuscript. Please ensure that you refer to Table 3 in your text; if accepted, production will need this reference to link the reader to the Table.

Thank you very much for your kind suggestion, and we apologize since there must have been a mistake in referencing tables. Now Table 3 is referenced in the first paragraph under Main analyses on line 8 of page 20 (under the section Main analyses). 

 

Responses to Reviewer #1’s comments: 

1. This study's authors develop a new experimental paradigm to examine the behavioral and neural correlates of episodic memory modulated by temporally delayed rewards. As the authors noted, it is novel to link reward-motivated memory encoding and temporal discounting together. This study's major findings are that the PFC activation is proportional to the degree of temporal discounting during the encoding and retrieval phase of the scene recognition memory task. In general, the writing of this manuscript is sound, and the data-analyses are comprehensive. I appreciated the authors' effort. 

Many thanks for your compliment.

2. However, the study is somewhat inconclusive. In particular, the behavioral results of this study are not clear and difficult to explain. Based on Figure 2 and its corresponding results, I did not see a real consistent data pattern. 

 Thank you for sharing your insightful concerns. We agree that the behavioral patterns shown here are not straightforward and lack statistical power to argue a robust effect of temporally delayed reward on memory. For clarification, we have modified the following: first, we decided to simplify the exploratory behavioral analyses by only report the “sooner (0 and 1 day)” vs. “later (7 and 28 days)” because although some are significant (7 days vs. 1 day and 28 days vs. 1 day), the comparisons between each delay do not contribute to telling a coherent story nor further neuroimaging analyses and only increases redundancy. This modification can be found on lines 10-16 of page 18 (under the section Behavioral results). Second, we made modifications in the abstract to not make strong conclusions from the behavioral trend (page 2 line 16). We hope these changes could make a conclusive story and alleviate your concerns regarding interpreting behavioral data.

3. The authors' explanation on the "later" (7 and 28 days) vs. "sooner" (0 and 1 day) contrast might be post hoc. 

We appreciate your sharp objection and agree with you that this may be of concern. However, although the differentiation between “sooner” and “later” might seem arbitrary or post hoc, we planned this division a priori based on previous studies on temporal discounting that grouped different timepoints into discrete bins, for the purpose of simplifying the covariates to make interpretations easier. One of the works we referred to was Ballard and Knutson (2009), one of the milestones of temporal discounting research, which binned continuous variables into discrete bins in order to form a factorial design: here, the monetary magnitude was binned into low ($10.00, $10.50), medium ($11.00, $13.00, $15.00), and high ($20.00, $25.00), and the temporal distance was also divided into low (0,7 days), medium (30, 60 days), and high delays (90, 180 days). The activation of volumes of interest during these conditions were extracted and submitted to a 2-way repeated-measures analysis of variance (ANOVA), with magnitude and delay as the within-subjects factors. In their supplementary material (Supplementary Fig. 2), they have justified the procedure of collapsing magnitude and delay into high, medium, and low groups by showing that there is a linear relationship between choice behavior and the factors (magnitude and delay). Since the structure of our intertemporal choice task is identical to that of Ballard and Knutson and the results have been replicated (as can be seen in the distribution of the participants’ k value), we believe it is also valid for us to assume a linear relationship between delay and the resulting subjective value of a reward option and collapsing the delays into bins. However, we acknowledge that performing the concatenation without justification may raise potential concerns. Therefore, in order to minimize potential concerns regarding the bin, we have added the rationale of concatenating the covariates in our manuscript, which is on lines 12-15 of page 18 (under the section Behavioral results). 

4. Since the behavioral results are not clear, I am not sure that the fMRI results really reflect the delayed reward's memory-enhanced effect. Instead, the fMRI results may merely reflect perceptual differences of temporal delays of different cues. Together, the inconclusive nature of data interpretation might lower the scientific value of this study.

Many thanks for making a keen point. We believe this could be an important potential concern for any study. However, we respectfully disagree with the idea that the behavioral and neural differences may have resulted from perceptual and/or stimulus-level differences of different conditions in this study. As seen in Fig 1 of our manuscript, the sole perceptual differences among different conditions in the cue phase of the encoding task are the numbers that represent the temporal delay. Should the differences in perceptual characteristics of the cue result in differences in the behavior of interest, this should be also applicable to any study that involves the standard intertemporal choice task used in Kable and Glimcher (2007) since this task also represents temporal delays in numbers. However, to the best of our knowledge, the concerns regarding perceptual differences of temporal delays or difference cues have not been raised. It is also unlikely that the perceptual differences of scene stimuli following the cue might have caused the behavioral and neural differences among conditions because the scene stimuli were randomly assigned to each condition. 

 

Responses to Reviewer #2’s comments: 

The authors utilized both behavioral and fMRI methods to investigate whether more immediate rewards are associated with better long-term memory. However, the findings were contrary to this hypothesis. They explained the unexpected results by the encoding-retrieval process, which is mainly represented by the LPFC activation. The experiments were well designed and conducted. The statistical analyses were reliable. The whole-manuscript was also well-written. I believe that this is a good study although the authors got findings contrary to their hypotheses.

We deeply thank you for your warm comments.

 

References

Ballard, K., & Knutson, B. (2009). Dissociable neural representations of future reward magnitude and delay during temporal discounting. Neuroimage, 45(1), 143-150.

Friedl-Werner, A., Brauns, K., Gunga, H. C., Kühn, S., & Stahn, A. C. (2020). Exercise-induced changes in brain activity during memory encoding and retrieval after long-term bed rest. NeuroImage, 223, 117359.

Kable, J. W., & Glimcher, P. W. (2007). The neural correlates of subjective value during intertemporal choice. Nature neuroscience, 10(12), 1625-1633.

Miotto, E. C., Balardin, J. B., Martin, M. D. G. M., Polanczyk, G. V., Savage, C. R., Miguel, E. C., & Batistuzzo, M. C. (2020). Effects of semantic categorization strategy training on episodic memory in children and adolescents. PloS one, 15(2), e0228866.

Mumford, J. A. (2012). A power calculation guide for fMRI studies. Social cognitive and affective neuroscience, 7(6), 738-742.

Murphy, K., & Garavan, H. (2004). An empirical investigation into the number of subjects required for an event-related fMRI study. Neuroimage, 22(2), 879-885.

Szucs, D., & Ioannidis, J. P. (2020). Sample size evolution in neuroimaging research: An evaluation of highly-cited studies (1990–2012) and of latest practices (2017–2018) in high-impact journals. NeuroImage, 221, 117164.

Xiao, X., Zhou, Y., Liu, J., Ye, Z., Yao, L., Zhang, J., ... & Xue, G. (2020). Individual-specific and shared representations during episodic memory encoding and retrieval. NeuroImage, 116909.

---

## [Decision Letter · Decision Letter 1]

16 Mar 2021

Neural correlates of episodic memory modulated by temporally delayed rewards

PONE-D-20-24115R1

Dear Dr. Han,

We’re pleased to inform you that your manuscript has been judged scientifically suitable for publication and will be formally accepted for publication once it meets all outstanding technical requirements.

Kind regards,

Kiyoshi Nakahara, PhD

Academic Editor

PLOS ONE

Reviewers' comments:

Reviewer's Responses to Questions

**Comments to the Author**

1. If the authors have adequately addressed your comments raised in a previous round of review and you feel that this manuscript is now acceptable for publication, you may indicate that here to bypass the “Comments to the Author” section, enter your conflict of interest statement in the “Confidential to Editor” section, and submit your "Accept" recommendation.

Reviewer #1: All comments have been addressed

Reviewer #2: All comments have been addressed

2. Is the manuscript technically sound, and do the data support the conclusions?

Reviewer #1: Yes

Reviewer #2: Yes

3. Has the statistical analysis been performed appropriately and rigorously? 

Reviewer #1: Yes

Reviewer #2: Yes

4. Have the authors made all data underlying the findings in their manuscript fully available?

Reviewer #1: Yes

Reviewer #2: Yes

5. Is the manuscript presented in an intelligible fashion and written in standard English?

Reviewer #1: Yes

Reviewer #2: Yes

6. Review Comments to the Author

Reviewer #1: In this revision, all my concerns have been resolved. Therefore, I recommend publishing this paper in PloS One.

Reviewer #2: (No Response)

7. PLOS authors have the option to publish the peer review history of their article (what does this mean?). If published, this will include your full peer review and any attached files.

Reviewer #1: No

Reviewer #2: No

---

## [Editor Report · Acceptance letter]

29 Mar 2021

PONE-D-20-24115R1 

Neural correlates of episodic memory modulated by temporally delayed rewards 

Dear Dr. Han:

I'm pleased to inform you that your manuscript has been deemed suitable for publication in PLOS ONE. Congratulations! Your manuscript is now with our production department. 

Kind regards, 

on behalf of

Dr. Kiyoshi Nakahara 

Academic Editor

PLOS ONE